# Trust Region Masking for Long-Horizon LLM Reinforcement Learning

Yingru Li [*] Jiacai Liu [* 1] Jiawei Xu [* 2] Yuxuan Tong [3] Ziniu Li [2] Baoxiang Wang [2 4]

## Abstract

Policy gradient methods for Large Language Models optimize a policy $\pi_\theta$ via a surrogate objective computed from samples of a rollout policy $\pi_{roll}$. However, modern LLM-RL pipelines suffer from unavoidable implementation divergences, such as backend discrepancies, Mixture-of-Experts routing discontinuities, and distributed training staleness. These factors cause an off-policy mismatch ($\pi_{roll} \neq \pi_\theta$), leading to approximation errors between the surrogate and the true objective. We demonstrate that classical trust region bounds on this error scale as $O(T^2)$ with sequence length $T$, rendering them vacuous for long-horizon tasks. To address this, we derive two new bounds: a Pinsker-Marginal bound scaling as $O(T^{1.5})$ and a Mixed bound scaling as $O(T)$. Crucially, both bounds depend on $D_{KL}^{max}$, the maximum token-level KL divergence across the sequence. As a sequence-level term, the divergence cannot be controlled by previous token-independent methods like PPO clipping. We propose Trust Region Masking (TRM), which masks entire sequences that violate the trust region. TRM reduces the vacuity of traditional bounds, offering a principled framework to mitigate training collapse in long-horizon LLM-RL.

## 1. Introduction

Reinforcement Learning (RL) has emerged as a cornerstone in training Large Language Models (LLMs) for complex tasks that demand extended reasoning, multi-step problem solving, and agentic behavior. As LLMs are deployed for long-horizon applications—ranging from mathematical reasoning (Zeng et al., 2025) and code generation (Liu et al., 2026) to autonomous tool use (Yang et al., 2024)—sequence lengths have rapidly expanded from hundreds to thousands of tokens. While policy gradient methods (Williams, 1992; Sutton et al., 1999), particularly Proximal Policy Optimization (PPO) (Schulman et al., 2017), remain the standard for these tasks, their theoretical foundations are increasingly strained by these extended horizons.

Trust region methods (Kakade & Langford, 2002; Schulman et al., 2015) offer a principled framework for policy optimization by utilizing a surrogate objective, $L(\pi_\theta)$, computed via samples from a rollout policy $\pi_{\text{roll}}$. The central appeal of this framework is the *monotonic improvement guarantee*: provided the surrogate objective improves and the policy remains within a specific trust region, the true objective $J(\pi_\theta)$ is guaranteed to increase. However, this guarantee is predicated on bounding the approximation error $|J(\pi_\theta) - J(\pi_{\text{roll}}) - L(\pi_\theta)|$, a quantity strictly dependent on the divergence between the rollout policy $\pi_{\text{roll}}$ and the training policy $\pi_\theta$.

In the context of modern LLM-RL systems, recent work demonstrates that off-policy mismatch ($\pi_{\text{roll}} \neq \pi_\theta$) is not merely an implementation snag but an inevitable consequence of trust-region methods (Liu et al., 2025; Yao et al., 2025). This mismatch arises from three primary sources:

- **Backend Discrepancies:** Discrepancies between high-throughput inference engines (e.g., vLLM (Kwon et al., 2023), SGLang (Zheng et al., 2024)) and precise training frameworks (e.g., Megatron-LM (Shoeybi et al., 2019), PyTorch FSDP (Zhao et al., 2023)) result in differing logits despite identical weights.

- **MoE Routing Discontinuities:** In Mixture-of-Experts models (Shazeer et al., 2017; Liu et al., 2024), minor numerical jitter can flip the expert selection, causing high-magnitude jumps in token probabilities that disrupt the smoothness assumptions of standard RL.

- **Distributed Staleness:** Asynchronous training pipelines (Espeholt et al., 2018; Nair et al., 2015) introduce latency between data generation and gradient updates, resulting in training occurring on $\pi_\theta$ while rollouts are generated by a stale weights $\pi_{\text{roll}}$.

We also provide a more detailed analysis of these mismatch sources in Appendix A.

[1]Fudan University [2]The Chinese University of Hong Kong, Shenzhen [3]ByteDance [4]Vector Institute. Correspondence to: Yingru Li <szrlee@gmail.com>.

*Proceedings of the 43rd International Conference on Machine Learning*, Seoul, South Korea. PMLR 306, 2026. Copyright 2026 by the author(s).

Given that $\pi_{\text{roll}} \neq \pi_\theta$ is inevitable, the magnitude of the approximation error becomes a critical concern. Crucially, classical error bounds (Kakade & Langford, 2002; Schulman et al., 2015) scale quadratically with sequence length ($O(T^2)$). For modern reasoning tasks where responses frequently exceed $T = 4096$ tokens, these bounds become theoretically vacuous. Even with a negligible per-token divergence of $D_{\text{KL}}^{\text{tok,max}} = 10^{-4}$, the classical bound predicts an error of $\approx 1677$—a value far exceeding any plausible reward improvement. Consequently, existing theory provides *no guarantee* that optimization steps in long-horizon LLM-RL actually improve performance.

To address this gap, we make the following contributions:

1. **Tighter Error Bounds:** We derive two novel bounds on the approximation error that significantly tighten theoretical guarantees: the *Pinsker-Marginal Bound* ($|\text{Error}| \leq \frac{4}{3}T^{3/2} \cdot D_{\text{KL}}^{\text{tok,max}}$, scaling as $O(T^{3/2})$) and the *Mixed Bound* ($|\text{Error}| \leq 2T \cdot \sqrt{D_{\text{KL}}^{\text{tok,max}} \cdot D_{\text{KL}}^{\text{seq}}}$, scaling as $O(T)$). These results improve upon classical $O(T^2)$ bounds by factors of $O(\sqrt{T})$ and $O(T)$, respectively (Section 4) .

2. **Analysis of Token-Level Failure:** We demonstrate that because both bounds depend on $D_{\text{KL}}^{\text{tok,max}}$—the maximum token-level divergence across the sequence—this error is inherently a sequence-level quantity. Consequently, it cannot be effectively controlled by token-independent interventions such as standard PPO clipping or token masking (Section 5).

3. **Trust Region Masking (TRM):** We propose TRM to mask entire sequences violating the trust region. This ensures $D_{\text{KL}}^{\text{tok,max}} \leq \delta$ for all accepted data, providing the first non-vacuous monotonic improvement guarantees for long-horizon LLM-RL (Section 6). Finally, we present empirical evidence demonstrating the training stability of TRM (Section 7).

## 2. Related Work

Reinforcement Learning (RL) has emerged as the standard paradigm for aligning Large Language Models (LLMs) in domains with verifiable outcomes, such as mathematical reasoning (Lightman et al., 2023; Zeng et al., 2025) and code generation (Gu, 2023; Liu et al., 2026). While policy gradient methods like Proximal Policy Optimization (PPO) (Schulman et al., 2017) and Group Relative Policy Optimization (GRPO) (Guo et al., 2025) are widely adopted due to their standard surrogate objectives, scaling them to long-horizon reasoning chains—where sequences often exceed thousands of tokens—introduces severe optimization and stability challenges. Consequently, recent efforts have focused on specialized scaling frameworks and alternative

optimization algorithms, including DAPO (Yu et al., 2026), MiniMax-M1 (Chen et al., 2025), and ScaleRL (Khatri et al., 2025). In this work, we focus specifically on PPO and GRPO, as they utilize the standard surrogate objective that our theoretical framework directly critiques.

Trust region methods, originating from Conservative Policy Iteration (CPI) (Kakade & Langford, 2002), provide the theoretical bedrock for stable RL training. TRPO (Schulman et al., 2015) practically applied these concepts by enforcing a KL divergence constraint between the training and rollout policies, guaranteeing monotonic improvement under the assumption that the divergence is small. However, the theoretical bounds underpinning these methods scale poorly with horizon length $T$. The classical result bounds the approximation error by $O(T^2)$ in the finite-horizon setting (Achiam et al., 2017). While acceptable for short-horizon control tasks ($T \approx 100$), these bounds become vacuous for modern LLMs where $T$ frequently exceeds 4000 tokens. Our work bridges this gap by deriving tight, non-vacuous bounds specifically for autoregressive sequence generation.

The divergence between the rollout and the training policy is a critical challenge in distributed RL. Frameworks like IMPALA (Espeholt et al., 2018) and APPO (Schulman et al., 2017) introduce correction mechanisms to mitigate staleness in actor-learner architectures. In the context of LLMs, this mismatch is exacerbated by the bifurcation of inference and training stacks. Rollout generation often utilizes high-throughput engines like vLLM (Kwon et al., 2023) or SGLang (Zheng et al., 2024), which may employ optimizations not present in the training loop (e.g., Megatron-LM (Shoeybi et al., 2019)).

Recent literature has characterized this implementation divergence or training-inference mismatch as a primary driver of RL training instability (Liu et al., 2025; Qi et al., 2025; Yao et al., 2025). To counteract these discrepancies, recent methods have proposed sequence-level rollout corrections and alternative grouping strategies, such as Group Sequence Policy Optimization (GSPO) (Zheng et al., 2025), alongside various masked importance sampling techniques (Qi et al., 2025). Unlike approaches that attempt to eliminate this mismatch engineering-wise, our framework treats the implementation divergence as an unavoidable constraint and derives theoretical bounds that account for it explicitly.

Standard PPO implementations in LLMs enforce trust regions via token-level clipping (Ziegler et al., 2019). However, autoregressive generation is inherently sequential; a small probability shift at an early token can lead to a vastly different semantic sequence, a phenomenon related to "exposure bias" (Bengio et al., 2015). Our proposed Trust Region Masking (TRM) addresses this by enforcing constraints at the sequence-level, ensuring that the theoretical preconditions for monotonic improvement are met in practice.

# 3. Background and Problem Setup

## 3.1. Autoregressive Language Generation

We focus on autoregressive language generation tasks where a policy $\pi_\theta$ generates a response $y = (y_1, \ldots, y_T)$ given a prompt $x$. Each token $y_t$ is sampled from a fixed vocabulary $\mathcal{V}$ according to:

$$y_t \sim \pi_\theta(\cdot \mid x, y_{<t}), \qquad (1)$$

where $y_{<t} = (y_1, \ldots, y_{t-1})$ represents the sequence of tokens generated prior to step $t$. The probability distribution for the full trajectory factorizes as:

$$P^{\pi_\theta}(y \mid x) = \prod_{t=1}^{T} \pi_\theta(y_t \mid x, y_{<t}). \qquad (2)$$

We define the *context* at step $t$ as $c_t = (x, y_{<t})$. The *context visitation distribution* induced by the policy $\pi$ is given by:

$$d_t^\pi(c_t) = P(x) \prod_{s=1}^{t-1} \pi(y_s \mid x, y_{<s}). \qquad (3)$$

This distribution represents the probability of reaching a specific context $c_t$ when following policy $\pi$.

## 3.2. The Optimization Problem

Given a scalar reward function $R(x, y) \in [0, 1]$, our objective is to maximize the expected reward:

$$J(\pi_\theta) = \mathbb{E}_{x \sim P(x)} \mathbb{E}_{y \sim \pi_\theta(\cdot|x)} [R(x, y)]. \qquad (4)$$

A fundamental challenge in this setting is the off-policy mismatch: we generate samples from a *rollout policy* $\pi_{\text{roll}}$, which generally differs from the *training policy* $\pi_\theta$. This necessitates the use of importance sampling or surrogate objectives to estimate gradients for $\pi_\theta$.

## 3.3. The Surrogate Objective

Following Kakade & Langford (2002) and Schulman et al. (2015), we utilize the surrogate objective:

$$L_{\pi_{\text{roll}}}(\pi_\theta) = \mathbb{E}_{\pi_{\text{roll}}} \left[ A \cdot \sum_{t=1}^{T} \rho_t \right], \qquad (5)$$

where $A = R(x, y) - b$ denotes the trajectory advantage (relative to a baseline $b$), and

$$\rho_t = \frac{\pi_\theta(y_t \mid c_t)}{\pi_{\text{roll}}(y_t \mid c_t)} \qquad (6)$$

is the per-token importance ratio. A critical property of this surrogate is that its gradient matches the true gradient at the reference policy (Kakade & Langford, 2002):

$$\nabla L_{\pi_{\text{roll}}}(\pi_\theta)\big|_{\pi_\theta = \pi_{\text{roll}}} = \nabla J(\pi_\theta)\big|_{\pi_\theta = \pi_{\text{roll}}}. \qquad (7)$$

While $L$ serves as a valid local approximation of $J$, the approximation error grows as the divergence between $\pi_\theta$ and $\pi_{\text{roll}}$ increases.

## 3.4. Divergence Measures

To quantify the discrepancy between policies, we employ the following divergence measures.

**Definition 3.1** (Token-level divergences). For a given context $c_t = (x, y_{<t})$, we define the Total Variation (TV) distance and the Kullback-Leibler (KL) divergence as:

$$D_{\text{TV}}^{\text{tok}}(c_t) := D_{\text{TV}}(\pi_\theta(\cdot|c_t) \| \pi_{\text{roll}}(\cdot|c_t))$$
$$= \frac{1}{2} \sum_v |\pi_\theta(v|c_t) - \pi_{\text{roll}}(v|c_t)|, \qquad (8)$$

$$D_{\text{KL}}(c_t) := D_{\text{KL}}(\pi_{\text{roll}}(\cdot|c_t) \| \pi_\theta(\cdot|c_t))$$
$$= \sum_v \pi_{\text{roll}}(v|c_t) \log \frac{\pi_{\text{roll}}(v|c_t)}{\pi_\theta(v|c_t)}. \qquad (9)$$

Consistent with TRPO (Schulman et al., 2015), we utilize the KL divergence from the rollout policy to the training policy, $D_{\text{KL}}(\pi_{\text{roll}} \| \pi_\theta)$. This direction is preferred because (1) it aligns with the constraint formulation in TRPO, and (2) it is computationally tractable using stored rollout logits.

Building on these token-level definitions, we define the corresponding sequence-level metrics:

**Definition 3.2** (Maximum and sequence-level divergences).

$$D_{\text{TV}}^{\text{tok,max}} := \max_{t, c_t} D_{\text{TV}}^{\text{tok}}(c_t), \qquad (10)$$

$$D_{\text{KL}}^{\text{tok,max}} := \max_{t, c_t} D_{\text{KL}}(c_t), \qquad (11)$$

$$D_{\text{KL}}^{\text{seq}} := D_{\text{KL}}(P^{\pi_{\text{roll}}}(\cdot|x) \| P^{\pi_\theta}(\cdot|x))$$
$$= \sum_{t=1}^{T} \mathbb{E}_{c_t \sim d_t^{\pi_{\text{roll}}}} [D_{\text{KL}}(c_t)]. \qquad (12)$$

The relationship between the sequence-level KL and the token-level KL is governed by the chain rule. We formally state and prove this property below.

**Lemma 3.3** (KL Chain Rule). *For any time step $t$, the divergence between context distributions decomposes as:*

$$D_{\text{KL}}(d_t^{\pi_{\text{roll}}} \| d_t^{\pi_\theta}) = \sum_{s=1}^{t-1} \mathbb{E}_{c_s \sim d_s^{\pi_{\text{roll}}}} [D_{\text{KL}}(c_s)]. \qquad (13)$$

*Proof.* The joint trajectory distribution factorizes as $P^\pi(x, y_{<t}) = P(x) \prod_{s=1}^{t-1} \pi(y_s|c_s)$. The KL divergence

is:

$$D_{\mathrm{KL}}(d_t^{\pi_{\mathrm{roll}}} \parallel d_t^{\pi_\theta}) = \mathbb{E}_{d_t^{\pi_{\mathrm{roll}}}} \left[ \sum_{s=1}^{t-1} \log \frac{\pi_{\mathrm{roll}}(y_s|c_s)}{\pi_\theta(y_s|c_s)} \right]$$

$$= \sum_{s=1}^{t-1} \mathbb{E}_{c_s \sim d_s^{\pi_{\mathrm{roll}}}}[D_{\mathrm{KL}}(c_s)]. \quad (14)$$

$\square$

Note that the definition of $D_{\mathrm{KL}}^{\mathrm{seq}}$ in Eq. (12) corresponds to the special case considering the full sequence.

Finally, we recall Pinsker's inequality (Pinsker, 1964), which bounds the Total Variation by the KL divergence:

$$(D_{\mathrm{TV}}^{\mathrm{tok}})^2 \leq \frac{1}{2} D_{\mathrm{KL}}. \quad (15)$$

Since the Total Variation distance is symmetric, Pinsker's inequality applies regardless of the KL direction. We use $D_{\mathrm{KL}}(\pi_{\mathrm{roll}} \parallel \pi_\theta)$ as our primary measure of divergence throughout this work.

## 4. Theoretical Analysis

We develop tighter error bounds for the surrogate objective. We firstly define the approximation error as:

$$\mathrm{Error}(\pi_\theta) := J(\pi_\theta) - J(\pi_{\mathrm{roll}}) - L(\pi_\theta). \quad (16)$$

This quantity measures the discrepancy between the true objective improvement and the surrogate improvement. Bounding $|\mathrm{Error}|$ is sufficient to guarantee that maximizing $L$ leads to a monotonic improvement in $J$.

This error can be decomposed into a sum over timesteps using the Performance Difference Identity (Kakade & Langford, 2002). Let the per-step advantage be $A_t^{\pi_{\mathrm{roll}}}(c, y_t) := \mathbb{E}_{\pi_{\mathrm{roll}}}[R \mid c, y_t] - \mathbb{E}_{\pi_{\mathrm{roll}}}[R \mid c]$, and define the expected advantage as $g_t(c_t) := \mathbb{E}_{y_t \sim \pi_\theta}[A_t^{\pi_{\mathrm{roll}}}(c_t, y_t)]$. The error is then given by:

$$\mathrm{Error} = \sum_{t=1}^{T} \left( \mathbb{E}_{c_t \sim d_t^{\pi_\theta}}[g_t(c_t)] - \mathbb{E}_{c_t \sim d_t^{\pi_{\mathrm{roll}}}}[g_t(c_t)] \right). \quad (17)$$

Intuitively, the error arises from evaluating the expected advantage $g_t$ under the mismatching context distribution $d_t^{\pi_{\mathrm{roll}}}$ rather than the true distribution $d_t^{\pi_\theta}$.

Our analysis relies on the following fundamental lemmas involving the advantage bound and context shift. We derive them formally below.

**Lemma 4.1** (Martingale Property). *For any context $c_t$, the expected advantage under the rollout policy is zero:* $\mathbb{E}_{y_t \sim \pi_{\mathrm{roll}}(\cdot|c_t)}[A_t(c_t, y_t)] = 0.$

*Proof.* By definition, $V(c_t) = \mathbb{E}_{y_t \sim \pi_{\mathrm{roll}}}[Q(c_t, y_t)]$ and $A_t = Q - V$. Thus, $\mathbb{E}_{\pi_{\mathrm{roll}}}[A_t] = \mathbb{E}_{\pi_{\mathrm{roll}}}[Q - V] = V - V = 0.$ $\square$

**Lemma 4.2** (Advantage Bound). *For rewards $R \in [0, 1]$, the expected advantage shift is bounded by $|g_t(c_t)| \leq 2D_{\mathrm{TV}}^{\mathrm{tok}}(c_t)$.*

*Proof.* Using the Martingale Property, we can rewrite $g_t$:

$$g_t(c_t) = \mathbb{E}_{\pi_\theta}[A_t] - \underbrace{\mathbb{E}_{\pi_{\mathrm{roll}}}[A_t]}_{=0}$$

$$= \sum_{y_t} (\pi_\theta(y_t|c_t) - \pi_{\mathrm{roll}}(y_t|c_t)) \cdot A_t(c_t, y_t). \quad (18)$$

Since rewards are in $[0, 1]$, $|A_t| \leq 1$. Applying Hölder's inequality:

$$|g_t(c_t)| \leq \sum_{y_t} |\pi_\theta(y_t|c_t) - \pi_{\mathrm{roll}}(y_t|c_t)| \cdot 1 = 2D_{\mathrm{TV}}^{\mathrm{tok}}(c_t).$$

$$(19)$$

$\square$

**Lemma 4.3** (Context Shift). *The context distribution shift accumulates linearly:* $\|d_t^{\pi_\theta} - d_t^{\pi_{\mathrm{roll}}}\|_{\mathrm{TV}} \leq (t-1) \cdot D_{\mathrm{TV}}^{\mathrm{tok,max}}.$

*Proof.* We proceed by induction. **Base case** ($t = 1$): $d_1^{\pi_\theta} = d_1^{\pi_{\mathrm{roll}}} = P(x)$, so the TV distance is 0. **Inductive step:** Using the coupling bound for product distributions:

$$\|d_{t+1}^{\pi_\theta} - d_{t+1}^{\pi_{\mathrm{roll}}}\|_{\mathrm{TV}} \leq \|d_t^{\pi_\theta} - d_t^{\pi_{\mathrm{roll}}}\|_{\mathrm{TV}} + D_{\mathrm{TV}}^{\mathrm{tok,max}} \quad (20)$$

$$\leq (t-1)D_{\mathrm{TV}}^{\mathrm{tok,max}} + D_{\mathrm{TV}}^{\mathrm{tok,max}}$$

$$= t \cdot D_{\mathrm{TV}}^{\mathrm{tok,max}}.$$

Similarly, by the KL chain rule (Eq. 12), the KL divergence satisfies $D_{\mathrm{KL}}(d_t^{\pi_{\mathrm{roll}}} \parallel d_t^{\pi_\theta}) \leq (t-1) \cdot D_{\mathrm{KL}}^{\mathrm{tok,max}}.$ $\square$

### 4.1. The Failure of Classical Bounds

The classical TRPO bound is derived by combining these lemmas via the inequality $|\mathbb{E}_P[f] - \mathbb{E}_Q[f]| \leq 2\|f\|_\infty \cdot D_{\mathrm{TV}}(P, Q)$. This yields:

$$|\mathrm{Error}| \leq 4(D_{\mathrm{TV}}^{\mathrm{tok,max}})^2 \sum_{t=1}^{T} (t-1)$$

$$= 2T(T-1)(D_{\mathrm{TV}}^{\mathrm{tok,max}})^2 \leq T(T-1) \cdot D_{\mathrm{KL}}^{\mathrm{tok,max}}.$$

$$(21)$$

This bound scales as $O(T^2)$. For a typical reasoning task with sequence length $T = 4096$ and a divergence of $D_{\mathrm{KL}}^{\mathrm{tok,max}} = 10^{-4}$, the bound yields $|\mathrm{Error}| \leq 1677$. Since the maximum possible reward is 1, a bound of 1677 is *vacuous*, offering no theoretical guarantee of improvement.

We are now ready to introduce two tighter bounds that significantly reduce this scaling.

## 4.2. The Pinsker-Marginal Bound

Our key insight is to apply Pinsker's inequality (Pinsker, 1964; Cover, 1999) to the *marginal* KL divergence rather than summing the per-step TV distances.

**Theorem 4.4** (Pinsker-Marginal Bound). *The approximation error is bounded by:*

$$|\text{Error}| \leq \frac{4}{3} T^{3/2} \cdot D_{\text{KL}}^{\text{tok,max}}. \tag{22}$$

*Proof.* From the chain rule, we have $D_{\text{KL}}(d_t^{\pi_{\text{roll}}} \| d_t^{\pi_\theta}) \leq (t-1) \cdot D_{\text{KL}}^{\text{tok,max}}$. Applying Pinsker's inequality to this marginal KL yields:

$$\|d_t^{\pi_\theta} - d_t^{\pi_{\text{roll}}}\|_{\text{TV}} \leq \sqrt{\frac{(t-1) \cdot D_{\text{KL}}^{\text{tok,max}}}{2}}. \tag{23}$$

Summing over $t$ and using $\sum_{k=0}^{T-1} \sqrt{k} \leq \frac{2}{3} T^{3/2}$:

$$\sum_{t=1}^{T} \|d_t^{\pi_\theta} - d_t^{\pi_{\text{roll}}}\|_{\text{TV}} \leq \sqrt{\frac{D_{\text{KL}}^{\text{tok,max}}}{2}} \cdot \frac{2}{3} T^{3/2}. \tag{24}$$

Using the advantage bound $\|g_t\|_\infty \leq 2\sqrt{D_{\text{KL}}^{\text{tok,max}}/2}$ (derived via Pinsker), we combine terms yielding:

$$|\text{Error}| \leq 2 \left( 2\sqrt{\frac{D_{\text{KL}}^{\text{tok,max}}}{2}} \right) \left( \sqrt{\frac{D_{\text{KL}}^{\text{tok,max}}}{2}} \cdot \frac{2}{3} T^{3/2} \right)$$

$$= \frac{4}{3} T^{3/2} \cdot D_{\text{KL}}^{\text{tok,max}}. \tag{25}$$

$\square$

For $T = 4096$, this bound yields $|\text{Error}| \leq 35.0$, a $48\times$ improvement over the classical result.

## 4.3. The Mixed Bound

We can also bound the TV shift uniformly using the total sequence-level KL divergence.

**Theorem 4.5** (Mixed Bound). *The approximation error is bounded by:*

$$|\text{Error}| \leq 2T \cdot \sqrt{D_{\text{KL}}^{\text{tok,max}} \cdot D_{\text{KL}}^{\text{seq}}}. \tag{26}$$

*Proof.* The marginal KL at any step $t$ is bounded by the full sequence KL: $D_{\text{KL}}(d_t^{\pi_{\text{roll}}} \| d_t^{\pi_\theta}) \leq D_{\text{KL}}^{\text{seq}}$. Applying Pinsker's inequality gives a uniform bound:

$$\|d_t^{\pi_\theta} - d_t^{\pi_{\text{roll}}}\|_{\text{TV}} \leq \sqrt{D_{\text{KL}}^{\text{seq}}/2}. \tag{27}$$

Summing over $T$ steps:

$$|\text{Error}| \leq \sum_{t=1}^{T} 2\|g_t\|_\infty \|d_t^{\pi_\theta} - d_t^{\pi_{\text{roll}}}\|_{\text{TV}}$$

$$\leq T \cdot 2 \left( 2\sqrt{\frac{D_{\text{KL}}^{\text{tok,max}}}{2}} \right) \sqrt{\frac{D_{\text{KL}}^{\text{seq}}}{2}}$$

$$= 2T \sqrt{D_{\text{KL}}^{\text{tok,max}} \cdot D_{\text{KL}}^{\text{seq}}}. \tag{28}$$

$\square$

This bound is strictly tighter when the divergence is sparse (i.e., $D_{\text{KL}}^{\text{seq}}$ is small relative to $T \cdot D_{\text{KL}}^{\text{tok,max}}$). For $D_{\text{KL}}^{\text{seq}} = 0.01$, this yields $|\text{Error}| \leq 8.2$, a $200\times$ tighter than TRPO.

## 4.4. Summary and Implications

We combine these results into a unified adaptive bound. Defining the minorizer $\mathcal{M}(\pi_\theta) := L(\pi_\theta) - |\text{Error}|_b$ with the bound error, monotonic improvement ($J(\pi_\theta) > J(\pi_{\text{roll}})$) is guaranteed if $\mathcal{M}(\pi_\theta) > 0$, where:

$$|\text{Error}|_b = \min \left\{ \frac{4}{3} T^{3/2} \cdot D_{\text{KL}}^{\text{tok,max}}, \ 2T\sqrt{D_{\text{KL}}^{\text{tok,max}} \cdot D_{\text{KL}}^{\text{seq}}} \right\}. \tag{29}$$

A numerical comparison of these bounds for a typical long-horizon scenario ($T = 4096$) is provided in Table 1. As shown, while the classical bound explodes to vacuous levels (1677.0), our Mixed bound remains tight (8.2), providing a usable theoretical signal.

*Table 1.* Comparison of error bounds for a long-horizon task ($T = 4096$) with $D_{\text{KL}}^{\text{tok,max}} = 10^{-4}$ and $D_{\text{KL}}^{\text{seq}} = 0.01$. All KL divergences use $D_{\text{KL}}(\pi_{\text{roll}} \| \pi_\theta)$.

| Bound | Formula | Scaling | Value ($T = 4096$) |
|---|---|---|---|
| Classical (TRPO) | $T(T-1)D_{\text{KL}}^{\text{tok,max}}$ | $O(T^2)$ | 1677.0 |
| **Pinsker-Marginal** | $\frac{4}{3} T^{3/2} D_{\text{KL}}^{\text{tok,max}}$ | $\mathbf{O(T^{3/2})}$ | **35.0** |
| **Mixed** | $2T\sqrt{D_{\text{KL}}^{\text{tok,max}} \cdot D_{\text{KL}}^{\text{seq}}}$ | $\mathbf{O(T)}$ | **8.2** |

Crucially, both bounds depend on $D_{\text{KL}}^{\text{tok,max}}$, the maximum token-level divergence. This confirms that the error is inherently a *sequence-level* quantity; controlling the average token KL is insufficient. We formalize the impossibility of removing this dependence in the following proposition.

**Proposition 4.6.** *There exists no function $f : \mathbb{R}^+ \to \mathbb{R}^+$ such that $D_{\text{KL}}^{\text{tok,max}} \leq f(D_{\text{KL}}^{\text{seq}})$ for all policy pairs.*

*Proof.* Consider a context $c^*$ that occurs with probability $\epsilon$ under the rollout distribution $d_t^{\pi_{\text{roll}}}$. Let the divergence be concentrated solely at the context: $D_{\text{KL}}(c^*) = 1$ and $D_{\text{KL}}(c) = 0$ for all $c \neq c^*$.

Then, the maximum token divergence is $D_{\text{KL}}^{\text{tok,max}} = 1$ (constant, independent of $\epsilon$). However, the sequence-level

divergence is $D_{\mathrm{KL}}^{\mathrm{seq}} = \epsilon \cdot 1 + (1-\epsilon) \cdot 0 = \epsilon$. As $\epsilon \to 0$, $D_{\mathrm{KL}}^{\mathrm{seq}} \to 0$ while $D_{\mathrm{KL}}^{\mathrm{tok,max}}$ remains 1. Thus, knowing only that $D_{\mathrm{KL}}^{\mathrm{seq}}$ is small provides no upper bound on $D_{\mathrm{KL}}^{\mathrm{tok,max}}$.
$\square$

This result necessitates our approach: sequence-level masking based on the *maximum* token divergence is required because token-level operation cannot control the worst-case error that drives training collapse.

## 5. Why Token-Level Methods Fail

Our theoretical analysis establishes that the approximation error is strictly bounded by $D_{\mathrm{KL}}^{\mathrm{tok,max}}$—a property of the *entire sequence*. In this section, we analyze why standard token-level interventions, such as PPO clipping or token masking, are mathematically insufficient to control this quantity, thereby failing to prevent optimization collapse in the presence of off-policy mismatch.

### 5.1. PPO Clipping and Gradient Leakage

PPO (Schulman et al., 2017) attempts to constrain updates via a clipped surrogate objective:

$$L^{\mathrm{CLIP}}(\pi_\theta) = \mathbb{E}\left[\sum_{t=1}^{T} \min\left(\rho_t A_t,\ \mathrm{clip}(\rho_t, 1-\epsilon, 1+\epsilon)A_t\right)\right].$$
(30)

While effective for standard control tasks, this mechanism fails when faced with the severe logit discrepancies common in LLMs. The failure mode is structural: the clipping operator $\mathrm{clip}(\cdot)$ is asymmetric. As detailed in Table 2, the mechanism provides safety when the policy attempts to increase the probability of a "good" action (Positive Advantage). However, it offers no protection against the penalization of "bad" actions (Negative Advantage) when the importance ratio $\rho_t$ is erroneously high (e.g., $\rho_t \gg 1+\epsilon$).

*Table 2.* PPO Clipping Analysis. When off-policy mismatch causes a spike in $\rho_t$ (e.g., due to MoE routing jitter), negative advantages result in **unbounded** gradients.

| Ratio $\rho_t$ | Advantage $A_t$ | Objective Value | Outcome |
|---|---|---|---|
| $> 1+\epsilon$ | $> 0$ | $(1+\epsilon)A_t$ | Clipped (Safe) |
| $< 1-\epsilon$ | $< 0$ | $(1-\epsilon)A_t$ | Clipped (Safe) |
| $> 1+\epsilon$ | $< 0$ | $\rho_t A_t$ | **Unclipped (Unbounded)** |
| $< 1-\epsilon$ | $> 0$ | $\rho_t A_t$ | **Unclipped (Unbounded)** |

In standard RL, this behavior is intended to penalize actions that are much more likely under the training policy but yield poor returns. However, in the context of mismatch in LLMs (e.g., implementation divergence), a large $\rho_t$ often represents numerical noise rather than a meaningful policy shift. Consequently, the optimizer receives a massive, erroneous gradient update that pushes the weights destructively, leading to the training collapse observed in practice.

### 5.2. The Insufficiency of Token Masking

A common heuristic to mitigate this is *token masking*: zeroing out the gradient contribution of specific tokens where $|\log \rho_t| > \delta$. The modified gradient becomes:

$$\nabla \approx \sum_{t=1}^{T} M_t \cdot \rho_t \nabla \log \pi_\theta(y_t|c) \cdot A,$$
(31)

where $M_t = 0$ if the divergence condition is met.

While this prevents immediate gradient explosion, it fails to satisfy the theoretical requirements for monotonic improvement. The error bounds derived in Section 4 depend on the divergence between the distributions $\pi_{\mathrm{roll}}$ and $\pi_\theta$ over the *entire trajectory*. Masking a specific token $t$ in the gradient computation does not alter the fact that the roll-out distribution $\pi_{\mathrm{roll}}$ has diverged from $\pi_\theta$. The quantity $D_{\mathrm{KL}}^{\mathrm{tok,max}}$ remains high, rendering the error bound vacuous. We formalize this limitation in the following proposition:

**Proposition 5.1.** *Token masking preserves vacuous bounds.* Let $\mathcal{T}_{\mathrm{mask}}$ be a token-level masking operator. The maximum token-level divergence of the underlying process remains unchanged:

$$D_{\mathrm{KL}}^{\mathrm{tok,max}}(\pi_\theta, \pi_{\mathrm{roll}}) = D_{\mathrm{KL}}^{\mathrm{tok,max}}(\mathcal{T}_{\mathrm{mask}}(\pi_\theta), \pi_{\mathrm{roll}}). \quad (32)$$

*Consequently, token masking alters the optimization direction but fails to restore the validity of the monotonic improvement guarantee.*

### 5.3. The Sequence-Level Imperative

This analysis reveals a fundamental dilemma for token-level methods, summarized in Table 3.

*Table 3.* Comparison of mitigation strategies

| Method | Gradient Leakage? | Theory Satisfied? |
|---|---|---|
| Standard PPO | Yes (Severe) | No |
| Token Masking | No | No ($D_{\mathrm{KL}}^{\mathrm{tok,max}}$ unchanged) |

The root cause is that the approximation error is cumulative and depends on the worst-case divergence in the sequence. If *any* token violates the trust region, the validity of the entire trajectory as an estimator for $J(\pi_\theta)$ is compromised. Therefore, rely on the tighter bounds derived in this work, we must exclude the *entire sequence* from the gradient computation, as proposed in Trust Region Masking (TRM).

## 6. Trust Region Masking

The theoretical analysis in Section 4 establishes that the approximation error is governed by $D_{\mathrm{KL}}^{\mathrm{tok,max}}$—a sequence-level quantity. Consequently, standard token-level interventions (such as PPO clipping) are insufficient to guarantee

monotonic improvement. To address this, we propose **Trust Region Masking (TRM)**, which masks *entire sequences* that violate the trust region constraints.

### 6.1. The Masked Surrogate Objective

We define a binary sequence mask $M(x, y) = \mathbb{I}[(x, y) \in \text{Trust Region}]$ and the corresponding masked surrogate objective:

$$L_{\text{masked}}(\pi_\theta) = \mathbb{E}_{\pi_{\text{roll}}} \left[ M(x, y) \cdot A(x, y) \cdot \sum_{t=1}^{T} \rho_t \right]. \quad (33)$$

The gradient is estimated using a batch of $N$ samples:

$$\nabla L_{\text{masked}} \approx \frac{1}{N} \sum_{i=1}^{N} M_i \cdot A^{(i)} \cdot \sum_{t=1}^{T} \rho_t^{(i)} \nabla \log \pi_\theta(y_t^{(i)} | c_t^{(i)}). \quad (34)$$

Crucially, the normalization factor is the *total* batch size $N$, not the count of accepted sequences. This ensures that rejected sequences effectively contribute zero gradient, preserving the unbiased nature of the estimate over the valid trust region. This acts as a rejection sampling mechanism: we simply choose not to learn from trajectories where the off-policy divergence renders the gradient unreliable.

### 6.2. Implementation and Divergence Estimation

**Exact KL Computation.** Following TRPO (Schulman et al., 2015), we utilize the forward KL divergence $D_{\text{KL}}(\pi_{\text{roll}} \| \pi_\theta)$. Because $\pi_{\text{roll}}$ logits are stored during data generation and $\pi_\theta$ logits are computed during the training forward pass, this quantity can be computed *exactly* without extra inference cost:

$$D_{\text{KL}}(c_t) = \sum_{v \in \mathcal{V}} \pi_{\text{roll}}(v | c_t) \log \frac{\pi_{\text{roll}}(v | c_t)}{\pi_\theta(v | c_t)}. \quad (35)$$

This eliminates the high variance associated with sample-based estimators used in standard PPO.

**Masking Criterion.** We employ a max-based criterion $M(x, y) = \mathbb{I}[\max_t D_{\text{KL}}(c_t) \leq \delta]$. This choice directly bounds $D_{\text{KL}}^{\text{tok,max}}$, ensuring the preconditions for our theoretical bounds are met. A key property of this criterion is *length-invariance*: unlike sum-based constraints, the threshold $\delta$ does not need to be adjusted as sequence length $T$ grows. In practice, to tolerate occasional outliers while maintaining robustness, one may combine this with an average-based constraint: $\frac{1}{T} \sum_t D_{\text{KL}}(c_t) \leq \delta_{\text{avg}}$.

**Sample-based Approximation.** In memory-constrained settings where storing full rollout logits is infeasible, one must rely on sample-based estimates derived from the importance ratio $\rho_t$. We recommend distinct criterion:

---

**Algorithm 1** Trust Region Masking (TRM)

---

**Require:** Threshold $\delta$; Batch $\mathcal{D} = \{(x^{(i)}, y^{(i)})\}_{i=1}^{N}$; Stored $\pi_{\text{roll}}$ logits

1: **Forward Pass:** Compute logits for $\pi_\theta$ on all data $(x, y) \in \mathcal{D}$
2: **for** each sequence $i \in \{1, \ldots, N\}$ **do**
3:     Compute per-token KL: $D_{\text{KL}}(c_t^{(i)}) = D_{\text{KL}}(\pi_{\text{roll}}(\cdot | c_t^{(i)}) \| \pi_\theta(\cdot | c_t^{(i)}))$
4:     Compute mask: $M_i \leftarrow \mathbb{I}\left[\max_t D_{\text{KL}}(c_t^{(i)}) \leq \delta\right]$
5: **end for**
6: **Backward Pass:** Compute $\nabla L_{\text{masked}}$ using only samples where $M_i = 1$
7: **Update:** $\theta \leftarrow \theta + \alpha \cdot \nabla L_{\text{masked}}$

---

1. **Max-Criterion ($k_2$):** We recommend the symmetric estimator $f(\rho) = \frac{1}{2}(\log \rho)^2$. This metric detects divergence symmetrically, flagging both support collapse ($\rho \to 0$) and impulse noise ($\rho \to \infty$) equally.

2. **Average-Criterion ($k_3$):** We recommend the estimator $f(\rho) = \rho - 1 - \log \rho$. This estimator is preferred because it is strictly non-negative and *unbiased* ($\mathbb{E}[f(\rho)] = D_{\text{KL}}$), ensuring that the sample average converges to the true sequence KL.

We provide a more detailed explanation in Appendix B.

### 6.3. Theoretical Guarantees

We formalize the properties of TRM in Theorem 6.1. By enforcing the trust region via rejection rather than penalty, TRM ensures the approximation error remains within the non-vacuous bounds derived in Section 4.

**Theorem 6.1** (TRM Core Properties). *Algorithm 1 with exact KL computation and threshold $\delta$ satisfies:*

1. *Bounded Divergence: For all accepted sequences (where $M = 1$), $D_{\text{KL}}^{\text{tok,max}} \leq \delta$.*

2. *Length-Invariant Threshold: The validity of the bound depends only on $\delta$, not sequence length $T$.*

3. *Bound Reduction Framework: For accepted sequences, the surrogate approximation error is guaranteed to remain restricted within the non-vacuous regime:*

$$|J(\pi_\theta) - J(\pi_{\text{roll}}) - L_{\text{masked}}| \leq \quad (36)$$
$$\min \left\{ \frac{4}{3} T^{3/2} \delta, \ 2T \sqrt{\delta \cdot D_{\text{KL}}^{\text{seq}}} \right\}.$$

*When this error envelope is smaller than the empirical surrogate improvement, it provides a mathematically rigorous justification for policy updates.*

*Proof.* **(1) Bounded Divergence:** By construction of Algorithm 1, $M_i = 1$ only if $\max_t D_{\mathrm{KL}}(c_t^{(i)}) \leq \delta$. Thus, for all accepted sequences, $D_{\mathrm{KL}}^{\mathrm{tok,max}} \leq \delta$.

**(2) Length-Invariant Threshold:** The masking criterion $\max_t D_{\mathrm{KL}}(c_t) \leq \delta$ depends only on the per-token maximum, not on any sum over $T$. Hence, $\delta$ is a fixed constant independent of sequence length.

**(3) Bound Reduction Framework:** For accepted sequences, $D_{\mathrm{KL}}^{\mathrm{tok,max}} \leq \delta$. Applying Theorems 4.4 and 4.5, the absolute deviation between true policy performance updates and the masked surrogate objective matches the stated minimum. While high-noise regimes can still push these values past standard reward boundaries, this reduction brings the theoretical evaluation out of vacuous $O(T^2)$ territories into standard optimization ranges. □

**Numerical Illustration** Revisiting the scenario from Table 1 ($T = 4096$, $\delta = 10^{-4}$), TRM ensures the error is bounded by **8.2** (Mixed bound) or **35.0** (Pinsker-Marginal). This contrasts sharply with the classical bound of **1677**, confirming that TRM provides the first theoretically grounded optimization path for long-horizon LLM reasoning.

## 7. Experiments

In this section, we provide empirical evidence validating the effectiveness of our TRM. We conduct experiments on mathematical reasoning using the vanilla Qwen3-8B-Base model under Zero-RL setup (Guo et al., 2025). The training dataset is a deduplicated version of DAPO-MATH-17k, and evaluation is performed on the AIME25 benchmark. We utilize GRPO (Shao et al., 2024) for advantage approximation with group size 16. The train batch size and rollout batch size are both set to 32, with a learning rate of $1 \times 10^{-6}$. For robust evaluation, we use sampling parameters Top-P $= 0.95$ and Temperature $= 1.0$, reporting the avg@32 score.

To simulate a realistic, high-throughput training environment, we introduce backend discrepancies by using vLLM for the inference (rollout) engine and PyTorch FSDP for the training engine. As discussed in Appendix A, the accumulation of floating-point differences between these backends acts as a primary source of off-policy divergence. To explicitly measure this mismatch between the rollout policy $\pi_{\mathrm{roll}}$ and the training policy $\pi_\theta$ during the update phase, we define the *Log Absolute Perplexity (PPL) Gap*. For a batch size $N$, this is calculated as:

$$
\Delta_{\mathrm{PPL}} = \frac{1}{N} \sum_{i=1}^{N} \left| \frac{1}{T_i} \sum_{t=1}^{T_i} \log \pi_\theta(y_t^{(i)} \mid c_t^{(i)}) - \right.
$$
$$
\left. \frac{1}{T_i} \sum_{t=1}^{T_i} \log \pi_{\mathrm{roll}}(y_t^{(i)} \mid c_t^{(i)}) \right|. \quad (37)
$$

This metric quantifies the average per-token log-probability drift attributable to implementation differences.

We compare our proposed TRM against the standard PPO Clipping baseline. For PPO Clipping, we adopt the range settings from DAPO (Yu et al., 2025), clipping ratios to $[0.8, 1.28]$. For the implementation of TRM, we evaluate two variants: TRM-Max, which masks sequences with threshold $\delta = 0.05$, and TRM-Avg, which masks sequences with threshold $\delta = 0.001$.

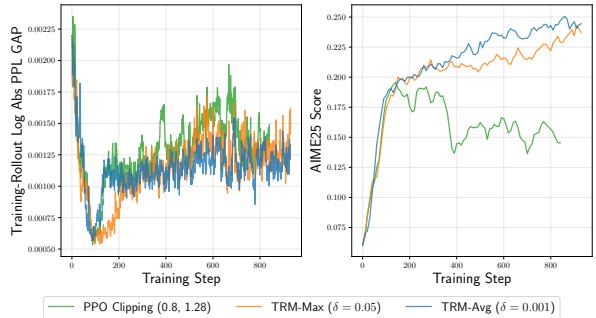

*Figure 1.* Comparison Results on TRM.

As illustrated in Figure 1, standard PPO Clipping fails to prevent collapse; the validation score degrades rapidly as the training progresses, correlating with an explosion in the PPL Gap. This confirms our theoretical finding in Section 5 that token-level clipping allows gradient leakage from mismatched trajectories. In contrast, both TRM variants maintain training stability. By rejecting entire sequences where implementation divergence exceeds the trust region, TRM keeps the PPL Gap bounded and ensures consistent improvement on the AIME25 benchmark.

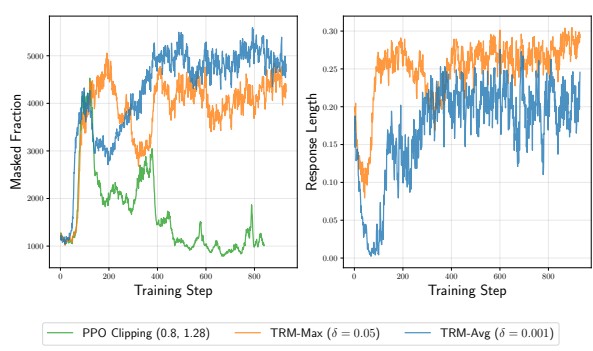

*Figure 2.* Trajectory evaluation metrics over time, detailing the average response length stability and the sequence masking rate for the TRM variants.

To rigorously evaluate the operational trade-offs and structural consequences of our hard-rejection mechanism, we track specific trajectory health metrics over the course of training in Figure 2. Crucially, the generated response lengths remain remarkably stable, plateauing consistently

near 5,000 tokens without any evidence of the premature length collapse common in long-horizon optimization. For the TRM-Max variant, this stabilization introduces a structural baseline masking rate of less than 30%, which inherently decreases raw token throughput. However, as demonstrated by the contrasting stability profiles, this sample reduction serves as a vital safety filter; while token-independent strategies like PPO clipping succumb to catastrophic optimization failures, TRM preserves absolute systemic stability.

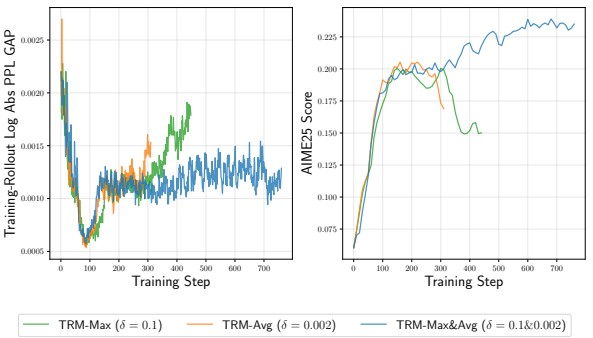

*Figure 3.* Effectiveness of Combined Criterion.

In practice, one can also apply both Max and Avg criteria simultaneously, which allows for the use of looser thresholds. As demonstrated in Figure 3, while TRM-Max (with $\delta = 0.1$) and TRM-Avg (with $\delta = 0.002$) individually fail to prevent training collapse due to the relaxed constraints, their combination (TRM-Max&Avg) successfully stabilizes training. This suggests that the two criteria are complementary: the max constraint catches extreme outliers, while the average constraint limits accumulated drift. Therefore, we recommend monitoring health metrics such as the Log Absolute PPL Gap to determine the appropriate criterion and thresholds when deploying TRM in real-world applications.

## 8. Conclusion and Discussion

Off-policy mismatch is unavoidable in modern LLM-RL due to implementation divergence. We show that classical trust region bounds, scaling as $O(T^2)$, become theoretically vacuous for long-horizon tasks common in reasoning domains. By deriving tighter Pinsker-Marginal bound ($O(T^{3/2})$) and the Mixed bound ($O(T)$) bounds, we establish that valid policy improvement depends strictly on the *maximum* token-level divergence—a quantity uncontrollable by standard token-level clipping. Consequently, we propose Trust Region Masking (TRM), a sequence-level intervention that masks entire sequences violating trust region constraints, thereby offering a principled framework to significantly reduce the vacuity of traditional bounds and ensure training stability for long-horizon LLM-RL. Our analysis extends beyond the specific algorithm proposed here, applying to

any method utilizing the standard surrogate objective, including REINFORCE and various PPO derivatives. The fundamental insight—that trust region constraints must be enforced at the sequence level rather than the token level—is universal for autoregressive generation.

While our exact formulation provides a rigorous framework to mitigate training collapse, its real-world implementation introduces notable trade-offs and limitations. Computing exact token-level KL from full rollout logits becomes computationally prohibitive and introduces a massive memory overhead for 70B+ models, meaning practitioners must rely on sample-based estimators ($k_2/k_3$) as pragmatic heuristics that do not strictly inherit our formal bounds. Furthermore, while our theory treats MoE routing jitter and distributed staleness as mathematically equivalent spikes in maximum token KL, our current empirical validation is scoped to backend discrepancies (vLLM vs. FSDP) as a stable proxy. Finally, TRM imposes a strict "rejection tax" on sample efficiency, explicitly sacrificing raw training throughput and altering the effective batch size to filter out unreliable gradients. Future work will focus on exploring soft masking via importance weighting, developing adaptive thresholding schedules to optimize this sample efficiency trade-off, and extending sequence-level trust regions to complex, multi-turn agentic workflows.

## Acknowledgements

Baoxiang Wang and Jiawei Xu are partially supported by the National Natural Science Foundation of China (72394361) and the Shenzhen Science and Technology Program (JCYJ20250604141218024, JCYJ20250604141032005).

## Impact Statement

This work is aim to advance the field of Reinforcement Learning for training Large Language Models. There are many potential societal consequences of our work, none which we feel must be specifically highlighted here.

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

# A. Details on Off-Policy Mismatch in LLM-RL

Prior work has established that off-policy mismatch ($\pi_{\text{roll}} \neq \pi_\theta$) is unavoidable in modern LLM-RL pipelines due to system-level constraints (Liu et al., 2025; Yao et al., 2025). Here, we detail the specific sources of this divergence.

## A.1. Backend Discrepancies

To maximize throughput, modern LLM infrastructures employ distinct software stacks for inference and training (Kwon et al., 2023; Zheng et al., 2024; Shoeybi et al., 2019). These differences are summarized below:

| Inference (vLLM/SGLang) | Training (Megatron/FSDP) |
| --- | --- |
| PagedAttention | FlashAttention-2 |
| FP8/INT8 KV-cache quantization | BF16/FP32 accumulation |
| Aggressive operator fusion | Tensor parallelism |

**Floating-point Non-associativity.** The root cause of divergence is the non-associativity of floating-point arithmetic: $(a \oplus b) \oplus c \neq a \oplus (b \oplus c)$. In attention mechanisms, the softmax denominator requires reducing over the context length. Different parallel reduction orders yield slightly different denominators. While these errors are negligible for a single token, they compound autoregressively over $T$ steps, leading to significant trajectory divergence.

## A.2. Mixture-of-Experts (MoE) Routing Discontinuities

In MoE architectures (Shazeer et al., 2017; Liu et al., 2024), the output is computed as:

$$y = \sum_{i \in \mathcal{K}} g_i(x) \cdot E_i(x), \quad \mathcal{K} = \text{Top-K}(h(x)), \tag{38}$$

where $h(x)$ represents the router logits. The Top-K operator is *discontinuous*. If numerical precision differences cause a shift $h_{\text{inf}} = h_{\text{train}} + \epsilon$ such that $|h_{(K)} - h_{(K+1)}| < \|\epsilon\|$, the set of selected experts $\mathcal{K}$ changes.

**Support Collapse.** A change in expert selection can drastically alter the output distribution. For instance, if $\pi_{\text{roll}}(\text{"apple"}) = 0.9$ but a routing flip causes $\pi_\theta(\text{"apple"}) \approx 0.001$, the importance ratio spikes to $\rho \approx 900$. This creates *impulse noise* in the gradient estimator, destabilizing training.

## A.3. Distributed Staleness

Large-scale training typically employs a decoupled actor-learner architecture (Espeholt et al., 2018; Nair et al., 2015):

- Actors generate rollouts using parameters $\theta_{\text{old}}$.

- Learner updates parameters to $\theta_{\text{new}}$.

- Latency introduces a lag of $k$ gradient steps between generation and consumption.

Consequently, $\theta_{\text{train}} = \theta_{\text{rollout}} + \sum_{i=1}^{k} \Delta\theta_i$, ensuring $\pi_{\text{roll}} \neq \pi_\theta$ even if implementations were identical.

**Summary.** These factors render off-policy mismatch *systemic* rather than incidental. Robust theoretical bounds must therefore account for $\pi_{\text{roll}} \neq \pi_\theta$ explicitly.

# B. Sample-Based Estimators ($k_2$ and $k_3$)

In settings where storing full logits is infeasible, we rely on sample-based estimators computed from the importance ratio $\rho_t = \pi_\theta(y_t|c_t)/\pi_{\text{roll}}(y_t|c_t)$. We analyze two estimators: $k_3$ (for unbiased averaging) and $k_2$ (for symmetric max-filtering).

### B.1. The $k_3$ Estimator for Averaging

The $k_3$ estimator is defined as $f(\rho) = \rho - 1 - \log \rho$.

- **Unbiased:** It is the only estimator where $\mathbb{E}_{y \sim \pi_{\mathrm{roll}}}[k_3(\rho)] = D_{\mathrm{KL}}(\pi_{\mathrm{roll}} \parallel \pi_\theta)$ exactly. This makes it ideal for the *average-based criterion* $\frac{1}{T} \sum k_3$, as the sample mean converges to the true sequence KL by the Law of Large Numbers.

- **Non-negative:** $k_3 \geq 0$ for all $\rho$, preventing cancellation artifacts common with simple log-ratios.

- **Asymmetric:** As shown in the table, $k_3$ penalizes $\rho \gg 1$ much more heavily than $\rho \ll 1$. This is correct for averaging (since high $\rho$ values are rare under $\pi_{\mathrm{roll}}$), but makes it poor for detecting single-token outliers.

### B.2. The $k_2$ Estimator for Max-Filtering

The $k_2$ estimator is defined as $f(\rho) = \frac{1}{2}(\log \rho)^2$. This is the second-order Taylor approximation of the KL divergence.

- **Symmetric:** $k_2(\rho) = k_2(1/\rho)$. It penalizes deviations in either direction equally.

- **Robustness:** For the *max-based criterion*, we require a detector that flags both "support collapse" (where $\pi_\theta \ll \pi_{\mathrm{roll}}$, $\rho \to 0$) and "impulse noise" (where $\pi_\theta \gg \pi_{\mathrm{roll}}$, $\rho \to \infty$). The $k_3$ estimator fails to flag $\rho \to 0$ aggressively (e.g., $k_3(0.01) \approx 3.6$), whereas $k_2$ treats it symmetrically to $\rho = 100$ ($k_2 \approx 10.6$).

- **Usage:** We recommend $k_2$ specifically for the max-threshold check: $M_i = \mathbb{I}[\max_t k_2(\rho_t) \leq \delta]$.

**Caveat.** Both $k_2$ and $k_3$ are single-sample approximations. While effective heuristics, the rigorous guarantees of Theorem 6.1 hold only when using the exact KL computed from full logits.

