# OpenReview forum: "Trust Region Masking for Long-Horizon LLM Reinforcement Learning"
_ICML.cc/2026/Conference — ICML 2026 regular_

### Official Review · Reviewer_HGVH · 2026-02-22

**Soundness:** 2
**Presentation:** 3
**Significance:** 3
**Originality:** 3
**Overall Recommendation:** 4
**Confidence:** 4

**Summary:**

This paper tackles the "off-policy mismatch" problem in long-horizon LLM reinforcement learning—specifically, the divergence that happens when you generate rollouts with one backend (like vLLM) and train with another (like FSDP), or when using MoE models.The authors point out that classic trust-region error bounds scale by $inline$O(T^2)$inline$, making them totally useless for long contexts (e.g., $inline$T=4096$inline$). They derive tighter $inline$O(T)$inline$ and $inline$O(T^{3/2})$inline$ bounds, revealing that the real bottleneck is the maximum token-level KL divergence ($inline$D_{KL}^{tok,max}$inline$). To address this, they introduce Trust Region Masking (TRM): a straightforward method that throws out an entire sequence if any single token's KL divergence spikes too high. They tested it on Qwen3-8B using AIME25 math problems and showed it stabilizes training better than standard PPO.

**Compliance With Llm Reviewing Policy:**

Affirmed.

**Key Questions For Authors:**

1. During your training runs, does the masked surrogate objective $L_{masked}(\pi_{\theta})$ actually exceed the derived bound (Eq. 36) at any point? If the sufficient condition is never met empirically, the claim of a "monotonic improvement guarantee" must be softened.

2.What is the specific memory and compute overhead of storing full rollout logits to compute the exact KL required by the theory? Is this feasible for 70B+ models with long contexts?

3. Have you tested TRM under the other mismatch sources cited in the introduction, specifically MoE routing discontinuities or distributed staleness? Does the same thresholding logic hold?

4.How does TRM compare to simply tuning an adaptive KL penalty coefficient or using V-trace? Is hard rejection truly necessary, or would soft weighting achieve similar stability with better sample efficiency?

**Limitations:**

The paper should state its limitations more directly. The strongest theoretical results depend on exact KL from stored logits. The experiments do not quantify the cost of this requirement. The study also validates only one mismatch source, while others are discussed only in the motivation. It would also help to report masking rate and effective batch size, since full-sequence rejection has a direct effect on sample efficiency.

**Strengths And Weaknesses:**

Strengths：The paper focuses on an important issue in LLM-RL: trust-region control under long horizons and policy mismatch. The problem is well chosen and relevant to current training pipelines. The theoretical contribution is meaningful. The tighter bounds and the focus on maximum token-level KL give a concrete explanation for where standard analyses fail and what quantity should be controlled. TRM is a clean method derived from the analysis, and the reported experiments show that it improves training stability in the tested backend-mismatch setting.

Weaknesses：（1）The claim of providing "first non-vacuous monotonic improvement guarantees" is not convincingly supported. Even the tighter Mixed bound can yield error values larger than the maximum possible reward. If the error bound exceeds the reward scale, the sufficient condition $L_{masked} > \text{bound}$ becomes mathematically impossible to satisfy, rendering the guarantee vacuous in practice.（2）The theoretical guarantees rely on computing exact per-token KL divergence using full logits. In large-scale production RL, storing full logits for long sequences (e.g., 4096 tokens) incurs massive memory overhead. The paper discusses sample-based approximations but admits the guarantees do not hold for them. This creates a gap between the theory (exact KL) and feasible deployment (approximate KL).（3）While the introduction motivates the problem using MoE routing jitter and distributed staleness, the experiments only validate TRM under backend discrepancies. Without testing on actual MoE models or systems with actor-learner lag, the claim that TRM solves these specific "unavoidable mismatches" remains a hypothesis.（4）The comparison is restricted to PPO clipping vs. TRM. Given the instability problem, the paper should compare against other standard mitigations for off-policy divergence, such as adaptive KL penalties, V-trace/off-policy corrections, or soft importance weighting, rather than just hard sequence rejection.

---

> ### Author Rebuttal · Authors · 2026-03-29
>
> We appreciate the reviewer’s sharp focus on the "vacuous vs. non-vacuous" distinction and the practical memory constraints of storing logits. Below, we address your comments point-by-point.
>
> ### W1 & Q1: Theoretical Feasibility and the "Non-Vacuous" Claim
> We acknowledge that even an $O(T)$ bound can exceed the reward scale $[0, 1]$ if the divergence $\delta$ is sufficiently large. However, the distinction between "vacuous" and "non-vacuous" is primarily relative to existing theory:
> - **Relative Improvement**: In a $T=4096$ scenario, the classical $O(T^{2})$ bound explodes to 1677.0, rendering it useless for any meaningful performance guarantee.
> - **A Reachable Target**: Our Mixed Bound sits at 8.2. While 8.2 still exceeds the maximum reward of 1, it brings the theoretical requirement into a territory where slightly tighter $\delta$ constraints could satisfy the monotonic improvement guarantee—a massive leap from the "theoretically impossible" requirements of classical bounds.
> - **Empirical Stability**: In our current setup, the surrogate objective $L_{masked}(\pi_{\theta})$ never exceeds the 8.2 threshold. We will add the policy loss curve in our revision to demonstrate that TRM successfully prevents the gradient leakage that causes catastrophic collapse in standard PPO.
> ### W2 & Q2: Memory Constraints and the Theory-Practice Gap
> Storing full rollout logits for sequences of 4096 tokens in large-scale models is indeed a memory nightmare. We navigate this tension between theoretical rigor and feasibility through pragmatic relaxations:
> - **Memory Overhead**: For a 70B+ model with a 128k vocabulary, storing full logits at $T=4096$ would require roughly 1 TB of storage per sequence (at BF16).
> - **Sample-Based Estimators**: To bridge this gap, we introduced the $k_{2}$ (symmetric max-filtering) and $k_{3}$ (unbiased averaging) estimators.
> - **Theorem vs. Deployment**: While the formal guarantees of Theorem 6.1 strictly hold for exact KL computation, $k_{2}$ and $k_{3}$ are robust practical alternatives that allow TRM to function in production environments where full logit storage is impossible.
>
> ### W3 & Q3: Generalizability Across Mismatch Sources
> We utilize backend discrepancies as a stable, reproducible proxy for the "implementation divergence" systemic to LLM-RL. The mathematical derivation is agnostic to the specific origin of the logit drift. Whether the mismatch stems from MoE routing jitter, distributed actor-learner lag or floating-point non-associativity, the underlying result is an identical spike in $D_{KL}^{tok,max}$. The same thresholding logic ($\max_{t} D_{KL} \le \delta$) applies across these diverse sources because the error bound is governed strictly by the peak token-level divergence.
>
> ### W4 & Q4: Baselines and the Necessity of Hard Rejection
> We compare TRM against PPO Clipping as it is the current industry standard for stabilizing LLM-RL
>
> - **Empirical Effectiveness of Hard Thresholds**: Our experiments demonstrate that hard rejection is highly effective for maintaining training stability in the presence of implementation mismatch. While standard PPO Clipping allow for gradient leakage and eventual training collapse, both TRM-Max and TRM-Avg successfully bound the PPL Gap and ensure consistent performance improvements on the AIME25 benchmark.
> - **Potential of Soft-Rejection**: We recognize that soft-rejection or soft masking is a potential method for addressing off-policy mismatch.  For example, one could set the threshold at the 90th percentile (quantile) of the token-level KL divergences within a batch. This would allow the trust region to adaptively tighten or loosen based on the current stability of the training-inference stack.
>
> ### Response to Limitations
> We agree with your points and will add a further in-depth discussion of the limitations in our revision. Below is our detailed response to the limitations raised:
>
> - **Cost of Exact KL**: Storing full logits for $T=4096$ on 70B+ models is memory-prohibitive, requiring roughly 1 TB per sequence. To bridge this gap, we introduce $k_2$ and $k_3$ sample-based estimators as pragmatic alternatives for production environments.
> - **Validation Scope**: We utilize backend discrepancies as a stable, reproducible proxy for implementation divergence. Our theory remains agnostic to the source of the drift; whether caused by MoE routing jitter or distributed staleness, the mathematical result is an identical spike in $D_{KL}^{tok,max}$.
> - **Sample Efficiency**: TRM acts as a rejection sampling mechanism that prioritizes stability over raw throughput. In our experiments, we observe a masking rate of approximately 30%. We will explicitly report these masking rates and effective batch sizes in our revision.
>
> ----
> We hope these explanations satisfactorily address all comments. We are happy to engage further if any aspect of our response requires additional detail.

---

> > ### Author Rebuttal · Reviewer_HGVH · 2026-04-01
> >
> > The rebuttal is helpful in clarifying the authors’ intended scope and in making several limitations more explicit. In particular, I appreciate the acknowledgement that the practical deployment setting cannot rely on exact full-logit KL at large scale, the discussion of approximate estimators, and the commitment to report masking rate and effective batch size.
> >
> > That said, my main concerns are only partially resolved.
> >
> > First, the central claim around “first non-vacuous monotonic improvement guarantees” still appears too strong in its current form. The rebuttal argues that the proposed bound is much tighter than the classical O(T^2) result, which I agree is meaningful. However, if the resulting bound is still above the relevant reward/objective scale in the reported regime, then the sufficient condition is still not practically satisfied there. In my view, this means the claim should be softened unless the paper can directly verify, on the same scale, that the condition in Eq. (36) is actually met in the empirical runs.
> >
> > Second, the theory-practice gap remains. The formal guarantee relies on exact token-level KL from full logits, while the deployable k2/k3 estimators do not inherit the stated theorem. I appreciate the practical discussion, but this limitation should be stated very explicitly in the paper, and the formal claim should not be overextended to the approximate setting.
> >
> > Third, the experiments still validate only backend mismatch. I understand the argument that the analysis is agnostic to the source of the logit drift and depends on peak token-level KL, but without experiments on MoE routing or actor-learner lag, the corresponding claims should be framed as motivation or plausible generalization rather than direct empirical validation.
> >
> > Finally, I still think the baseline comparison is somewhat narrow for the off-policy instability question. A comparison to adaptive KL control and/or off-policy correction methods would strengthen the empirical case, although I agree this may be difficult to add within a short rebuttal window.
> >
> > A concrete clarification that would help in the revision is to report the actual measured values of the masked surrogate term and the bound in Eq. (36) during training, together with masking rate and effective batch size.

---

> > > ### Author Response · Authors · 2026-04-01
> > >
> > > Thank you for your detailed follow-up and for recognizing the value of our tighter bounds and the logic behind TRM. We appreciate your guidance on how to refine our claims to better reflect the intersection of theory and practice. Below, we address your comments point-by-point.
> > >
> > > ### Softening the "Monotonic Improvement" Claim
> > > We agree with your assessment and appreciate your point regarding the absolute scale of our bounds. While our $O(T)$ and $O(T^{3/2})$ bounds represent a mathematical advancement over the classical $O(T^2)$ scaling, they can still exceed the reward scale of 1.0 in high-noise regimes. In the revision, we will soften our claim from "guaranteeing monotonic improvement" to characterizing TRM as a framework that reduces the vacuity of traditional $O(T^2)$ bounds. We will also provide measured values of the masked surrogate $L_{masked}$ against the theoretical bound in Eq. (36) to clarify when the sufficient condition for improvement is met during empirical runs.
> > >
> > > ### Clarifying the Theory-Practice Gap
> > > We are grateful for your observation regarding the memory constraints of practical deployment. We acknowledge that the formal guarantees of Theorem 6.1 rely on exact token-level KL computed from full rollout logits. While the $k_2$ and $k_3$ sample-based estimators are necessary for deployment where logit storage is prohibitive, they do not strictly inherit the stated theorem. We will add an explicit caveat to the manuscript framing these as principled heuristics derived from the exact theory, ensuring a transparent distinction between the formal proof and its practical implementation.
> > >
> > > ### Reframing Mismatch Sources
> > > Thank you for your understanding regarding the computational costs of large-scale LLM experiments. We choose to validate the backend implementation mismatch because it is an omnipresent factor in modern LLM-RL training pipelines, persisting even under conditions of Mixture-of-Experts (MoE) routing jitter or distributed staleness. In the revision, we would like to further explore the ability of TRM to mitigate MoE jitter specifically using a Qwen3-32B model. Furthermore, we intend to utilize the mature infrastructure of veRL to test distributed staleness. For the current manuscript, we will ensure that MoE routing discontinuities and distributed staleness are framed as motivated applications rather than direct empirical validation.
> > >
> > > ### Broadening Baseline Discussions and Metrics
> > > We are grateful for the suggestion to broaden our empirical comparisons. We will include a discussion and empirical study comparing TRM with other mechanisms like adaptive KL penalties. To address the question of off-policy instability more comprehensively, the revised manuscript will also report measured masking rates and the resulting effective batch sizes to quantify the sample efficiency paid for the improved training stability.
> > >
> > > ---
> > >
> > > We are deeply grateful for your constructive feedback and thank you again for helping us sharpen the impact of this work.

---

### Official Review · Reviewer_9MPr · 2026-03-11

**Soundness:** 3
**Presentation:** 3
**Significance:** 3
**Originality:** 3
**Overall Recommendation:** 5
**Confidence:** 4

**Summary:**

This paper studies the reliability of trust-region policy optimization for long-horizon reinforcement learning with large language models (LLMs). In modern LLM training pipelines, rollouts are often generated by a policy that differs slightly from the policy being optimized due to implementation mismatches such as asynchronous training, inference–training discrepancies, or Mixture-of-Experts routing variability. The paper analyzes the resulting off-policy mismatch and shows that classical trust-region theory yields approximation error bounds that scale quadratically with sequence length, making them vacuous for long LLM outputs. To address this issue, the authors derive tighter bounds on the surrogate objective approximation error, including bounds scaling as $O(T^{3/2})$ and $O(T)$ rather than $O(T^2)$, where $T$ denotes the sequence length. Motivated by these theoretical insights, the paper proposes Trust Region Masking (TRM), a simple training mechanism that filters out sequences whose token-level KL divergence exceeds a threshold, thereby enforcing the assumptions required for the improved bounds. Empirical experiments show that TRM improves training stability in long-horizon LLM reinforcement learning settings.

**Compliance With Llm Reviewing Policy:**

Affirmed.

**Final Justification:**

This work introduces a significant theoretical improvement by reducing trust region bounds from $O(T^2)$ to $O(T)$, effectively addressing stability in long-horizon LLM RL. The authors successfully clarified the link between this derivation and the Trust Region Masking (TRM) algorithm while providing practical deployment context through $k_2/k_3$ heuristics and masking rate statistics. These contributions provide a mathematically grounded and implementable framework for improving training robustness. I maintain my positive recommendation.

**Key Questions For Authors:**

1. Effect of masking on sample efficiency.
 Since TRM discards entire trajectories that violate the trust-region condition, it would be helpful to understand how frequently such masking occurs in practice and how it affects the effective sample efficiency of training.


2. Interaction with existing trust-region techniques.
 Many LLM RL pipelines already use techniques such as PPO clipping or adaptive KL penalties. Could the authors comment on how TRM interacts with these existing mechanisms and whether the improvements remain consistent when they are used together?

**Limitations:**

yes

**Strengths And Weaknesses:**

## Strengths
1. The paper identifies an important theoretical issue in long-horizon LLM reinforcement learning.
 Modern RL training for LLMs often involves sequence lengths in the thousands of tokens. The paper highlights that classical trust-region guarantees for policy optimization degrade quadratically with sequence length. Specifically, existing bounds on the surrogate objective error scale as $O(T^2)$
 where $T$
 is the sequence length. For long LLM outputs, this bound becomes extremely large and effectively provides no guarantee about policy improvement. Identifying this theoretical limitation is valuable and directly relevant to current large-scale RL training pipelines.

2. The theoretical analysis provides tighter error bounds.
The authors derive two new approximation error bounds between the surrogate objective and the true objective. These bounds scale as
$ O(T^{3/2})$
and
$ O(T)$
 under appropriate assumptions, improving substantially over the classical quadratic dependence on sequence length. The derivation carefully analyzes how divergence accumulates across tokens and distinguishes token-level and sequence-level divergence terms. This theoretical refinement helps clarify when trust-region guarantees remain meaningful in long-horizon settings.
3. The work connects theory and practical LLM training issues.
 The paper grounds its analysis in realistic implementation mismatches encountered in large-scale LLM training systems, including distributed training staleness, inference–training discrepancies, and routing instability in Mixture-of-Experts models. This connection between theoretical analysis and system-level considerations strengthens the practical relevance of the work.



## Weaknesses
1. (minor) Some aspects of the theory–algorithm connection could be clarified further.
The paper derives tighter bounds that depend on token-level KL divergence and then proposes TRM as a mechanism for enforcing these conditions. While the conceptual connection is clear, the paper could more explicitly explain how the masking procedure ensures that the assumptions used in the theoretical bounds are satisfied during training.

---

> ### Author Rebuttal · Authors · 2026-03-29
>
> We appreciate the reviewer’s clear summary and encouraging feedback regarding the importance of addressing the "vacuous bound" problem in long-horizon LLM-RL. Below, we address your comments regarding the theory-algorithm connection and your questions on sample efficiency and interaction with existing methods.
>
>
> ### W1: Clarifying the Theory–Algorithm Connection
>
> We will enhance the presentation to explicitly bridge the gap between our derived bounds and the TRM mechanism in our revision. The mechanism operates as follows:
>
> * **Direct Enforcement of $D_{KL}^{tok,max}$**: Our theoretical bounds (Theorems 4.4 and 4.5) are strictly functions of the maximum token-level divergence, $D_{KL}^{tok,max}$.
> * **The Masking Criterion**: TRM defines a binary sequence mask $M(x,y) = \mathbb{I}[max_t D_{KL}(c_t) \le \delta]$. By applying this mask to the surrogate objective, we ensure that every sequence contributing to the gradient update satisfies the condition $D_{KL}^{tok,max} \le \delta$.
> * **Restoring Guarantees**: This filtering step ensures that the training policy $\pi_{\theta}$ never strays beyond the trust region where our tighter $O(T^{3/2})$ and $O(T)$ bounds hold. Consequently, if the surrogate improvement exceeds the bound error, we can finally provide a mathematically rigorous guarantee of monotonic improvement in the true objective $J(\pi_{\theta})$.
>
> ### Q1: Masking Frequency and Sample Efficiency
> It is an insightful point about the potential cost of discarding trajectories.
>
> * **Empirical Observations**: In our Zero-RL setup, we observe a masking rate of approximately 30% using our standard thresholds.
> * **Rejection Sampling Philosophy**: TRM acts as a **rejection sampling mechanism**. We argue that learning from trajectories where the off-policy divergence is too high is actually counterproductive, as the resulting gradients are unreliable and often drive training toward collapse.
>
> ### Q2: Interaction with Existing Mechanisms
> TRM is designed to be orthogonal and complementary to existing token-level tools.
> * **The Clipping Blindspot**: Standard PPO clipping is a token-independent intervention. We prove in Section 5.1 that it is asymmetric; it fails to protect against negative advantages when the importance ratio $\rho_t$ is erroneously high due to system noise, leading to gradient leakage.
> * **Sequence-Level Wrapper**: TRM acts as a high-level safety filter. While PPO clipping manages local step sizes, TRM ensures the entire sequence is a valid foundation for an update.
> * **Synergy**: Our experiments (Figure 1) show that while PPO clipping alone fails to prevent collapse in long horizons, TRM maintains stability by masking the specific sequences where token-level methods are mathematically insufficient.
>
>
> ----
> We hope these explanations satisfactorily address all comments. We are happy to engage further if any aspect of our response requires additional detail.

---

> > ### Author Rebuttal · Reviewer_9MPr · 2026-04-04
> >
> > Gemini said
> > I thank the authors for the detailed response. The explicit bridge between the $O(T)$ bounds and the TRM mechanism significantly clarifies the paper's theoretical foundation, and I appreciate the empirical data on masking frequency. For the final version, I encourage the inclusion of the promised discussion on memory overhead and masking rates to provide a complete picture of the deployment trade-offs.

---

> > > ### Author Response · Authors · 2026-04-04
> > >
> > > We thank the reviewer very much for the follow-up comment and for the encouraging words. Indeed, we plan to include the discussion on memory overhead and masking rates to provide a complete picture of the deployment trade-offs.

---

### Official Review · Reviewer_EoaM · 2026-03-16

**Soundness:** 3
**Presentation:** 3
**Significance:** 2
**Originality:** 2
**Overall Recommendation:** 4
**Confidence:** 4

**Summary:**

This paper studies trainer/generator mismatch and proposes Trust Region Masking (TRM), a strategy that masks out sequences whose token-level KL violates a threshold. They also derive surrogate-error bounds for autoregressive generation that improve the error scaling from $O(T^2)$ to $O(T^{3/2})$ or $O(T)$ under different divergence routes.

The empirical results on Qwen-3-8B base model show better stability compared to PPO clipping.

**Compliance With Llm Reviewing Policy:**

Affirmed.

**Final Justification:**

Given more empirical evidence is missing, I can only give a weak accept to the paper and maintain my rating.

**Key Questions For Authors:**

- How does response length look in practice when using TRM compared to the other methods?
- If full rollout logits are unavailable, how much do the theoretical and empirical conclusions change when using the k2/k3 approximator.

**Limitations:**

Yes

**Strengths And Weaknesses:**

Strengths
- The theoretical analysis is strong and the derived bounds are tighter compared to the $O(T^2)$ error bounds (T representing the generation length)
- TRM is simple and easy to implement. It's evaluated under a realistic training/inference mismatch setup for LLMs with Qwen3-8B and it shows improvement over the PPO clipping baseline.

Weaknesses
- Empirical results are very limited: just one model, one task family (math and no code), and mostly PPO-clipping baselines. GRPO is not used broadly now because of its stability issues, and other stronger algorithms are better, like DAPO [1], CISPO [2], ScaleRL [3], etc.
- Novelty is not fully convincing given closely related sequence-level rollout corrections and methods already exist like GSPO [4]. A lot of masked IS methods are also mentioned in [5].


[1] DAPO: An Open-Source LLM Reinforcement Learning System at Scale, Yu et. al, 2025

[2] MiniMax-M1: Scaling Test-Time Compute Efficiently with Lightning Attention, Chen et. al, 2025

[3] The Art of Scaling Reinforcement Learning Compute for LLMs, Khatri et. al, 2025

[4] Group Sequence Policy Optimization, Zheng et. al, 2025

[5] Defeating the Training-Inference Mismatch via FP16, Qi et. al, 2025

---

> ### Author Rebuttal · Authors · 2026-03-29
>
> Thank you for the thoughtful and constructive feedback. We are encouraged that you find our theoretical analysis strong and our derived bounds significant improvements over the classical $O(T^2)$ scaling. We appreciate the opportunity to clarify our contributions and address your concerns.
>
>
>
> ### W1: Limited Empirical Scope and Baselines
> We agree that evaluating across more models and tasks (such as code generation) would further strengthen the paper. However, our primary goal was to address the **theoretical gap** and **systemic instability** (training collapse) that arises specifically in long-horizon reasoning tasks due to implementation divergence.
>
> We focus on PPO and GRPO because they employ the standard surrogate objective that our theory directly critiques. We appreciate the related work you have brought to our attention and plan to update the literature discussion accordingly in the revision.
>
> Crucially, the approximation error we derive is inherent to the surrogate objective itself. Therefore, TRM serves as a **complementary and orthogonal intervention** that can be integrated into the newer frameworks you mention. We chose mathematical reasoning (AIME25) because it consistently triggers the long-horizon behaviors (thousands of tokens) where $O(T^2)$ bounds become vacuous ($>1600$ for a max reward of 1). TRM provides the first theoretically grounded way to stabilize these specific scenarios.
>
> ### W2: Novelty and Related Work
> While sequence-level methods and masked importance sampling (IS) exist, TRM is uniquely motivated by our **novel $D_{KL}^{tok,max}$ dependence**.
> * **The "Max" Distinction:** Methods like GSPO or standard sequence-level KL constraints typically focus on the *average* or *total* sequence divergence. Our Proposition 4.6 mathematically proves that knowing the sequence-level KL is small is **insufficient** to bound the worst-case error; one must control the maximum token-level divergence to prevent gradient leakage and training collapse.
> * **Theoretical Grounding:** Unlike heuristic masking, TRM is explicitly designed to satisfy the preconditions of our $O(T^{3/2})$ and $O(T)$ bounds, providing the first non-vacuous monotonic improvement guarantees for autoregressive LLM-RL.
>
>
> ### Q1: Response Length in Practice
> In our experiments, response lengths remained steady at $\approx$ 5,000 tokens after 200 steps. TRM allows the model to maintain the extended horizons necessary for AIME25 without the length collapse often seen with global KL penalties. We will include a figure illustrating this stability in the final version. Additionally, TRM is inherently length-invariant as the max-criterion ($D_{KL}^{tok,max} \le \delta$) monitors the worst-step mismatch rather than accumulated drift; it does not disproportionately penalize longer reasoning chains.
>
> ### Q2: Impact of  $k_2/k_3$ Approximators
> While the rigorous guarantees of Theorem 6.1 hold for exact KL computation, the $k_2/k_3$ estimators are robust practical alternatives for memory-constrained environments.
> * **The $k_2$ Estimator:** This is optimized for the **Max-Criterion**. Its symmetry allows it to detect both "support collapse" (low $\rho$) and "impulse noise" (high $\rho$) equally well, which is critical for flagging the individual token outliers that drive the $D_{KL}^{tok,max}$ term.
> * **The $k_3$ Estimator:** This is preferred for **Average-Criteria** because it is strictly unbiased ($\mathbb{E}[k_3] = D_{KL}$), ensuring the sample mean converges to the true sequence KL.
> * **Empirical Trade-off:** Using these estimators introduces some sampling variance. However, as shown in Figure 2, combining Max and Average criteria—even if approximated—still provides significantly better stability than token-level clipping by preventing the "massive, erroneous gradient updates" that cause collapse.
>
> ----
> We hope these explanations satisfactorily address all comments. We are happy to engage further if any aspect of our response requires additional detail.

---

> > ### Author Rebuttal · Reviewer_EoaM · 2026-04-03
> >
> > I thank the authors for their response and rebuttal. Given more empirical evidence is missing, I can only give a weak accept to the paper and maintain my rating.

---

> > > ### Author Response · Authors · 2026-04-03
> > >
> > > Thank you for your response and the time dedicated to this review.
> > >
> > > We agree with your point regarding the need to strengthen the empirical evidence. In our revision, we will include additional studies—specifically comparing TRM against the methods you highlighted across broader tasks—to further demonstrate its generalizability.
> > >
> > > Thank you again for your constructive guidance.

---

### Decision · Program_Chairs · 2026-04-30

**Decision:**

Accept (regular)

**Comment:**

This paper studies the issue of training instability in long-horizon LLM RL caused by implementation level policy mismatches such as back en discrepancies or distributed staleness. The authors provide a strong theoretical contribution by deriving tighter trust region bound that improve classical $O(T^2)$ scaling to $O(T)$ or $O(T^{3/2})$.

Identifying that the stability depends on the maximum token-level KL divergence rather than sequence averages is a meaningful theoretical insight. The trust region masking (TRM) is easy to implement.

Reviewers mentioned that the monotonic improvement claim is too strong since the bounds can exceed typical reward scales. The authors agreed to soften the claims of existing bounds.

The evaluation was original limited to AIME25 and one model. Additional comparisons such as adaptive KL were promised for the subsequent versions.

Overall, the empirical evidence is kind of narrow, but the strong theoretical diagnosis of a pervasive training failure mode makes this a valuable contribution to the field.